# Genomic characteristics of *Salmonella enterica* serovar Blockley

Hidemasa Izumiya,[1] Chien-Shun Chiou,[2] Masatomo Morita,[1] Toshio Sato,[3] Akio Noguchi,[3] Tetsuya Harada,[4] Yukihiro Akeda,[1] Makoto Ohnishi[1]

**ABSTRACT**   Non-typhoidal *Salmonella* (NTS) is a significant cause of foodborne illness worldwide, with increasing antimicrobial resistance posing a public health concern. *Salmonella enterica* serovar Blockley (*S*. Blockley) is relatively uncommon, and its antimicrobial resistance profile and population structure have been understudied. This study presents a comprehensive genomic analysis of 264 *S*. Blockley isolates from diverse geographical regions to elucidate antimicrobial resistance patterns and population structure. Bayesian analysis classified these genomes into 10 distinct groups (BAPS A to BAPS J), further categorized into two lineages, R and S. Lineage R comprised six BAPS clusters (BAPSs A–F), predominantly found in Asia and Africa, all of which harbored the azithromycin resistance gene *mph(A)* and other resistance determinants. In contrast, lineage S, lacking *mph(A)*, comprised the remaining four BAPS clusters, which were primarily found in Europe and the Americas. Several types of mutations in *gyrA* were found in lineage R, which were specific to BAPS clusters. These BAPS clusters exhibited distinct geographic distributions, with BAPS B, BAPS D, and BAPS E unique to China, Taiwan, and Japan, respectively, while BAPS H and BAPS I were predominantly found in the United States. Temporal phylogenetic analysis suggested that lineage R diverged in the 1980s, with notable microevolutionary changes. The presence of a genomic island with *mph(A)*, *aph(3')-Ia*, *aph(3")-Ib*, *aph(6)-Id*, and *tet(A)* in lineage R underscores the public health threat, highlighting a need for continuous surveillance.

**IMPORTANCE**   Antimicrobial resistance in *Salmonella* is a global public health concern. In this study, we focused on serovar Blockley, and a whole-genome analysis revealed its global population structure. The results revealed the existence of azithromycin-resistant strains, which were characterized both phylogenetically and geographically. The resistance genes were transmitted via genomic islands, and their micro-scale evolution was also revealed. Our findings are the first to reveal the dissemination of antimicrobial resistance genes, including azithromycin, in serovar Blockley, and provide valuable insights into understanding the spread of antimicrobial resistance.

**KEYWORDS**   *Salmonella*, Blockley, genome analysis, multidrug resistance, drug resistance evolution, azithromycin

$S$almonella is a Gram-negative rod that causes major foodborne infections worldwide. Nontyphoidal *Salmonella* (NTS) is a major contributor to the global burden of diarrheal disease in humans, with an estimated 197.35 million episodes and 84,799 deaths annually (1).

In the European Union, salmonellosis is the second most common zoonotic disease in humans, with 65,208 cases reported in 2022 (2). Serovar Enteritidis was the most prevalent, accounting for 67.3% of cases, followed by serovar Typhimurium (13.1%) and monophasic Typhimurium variant I, 1,4,[5],12:i:- (4.3%). NTS typically causes gastroenteritis and is self-limiting, but antimicrobial therapy is necessary for extraintestinal infections. Antimicrobial resistance in *Salmonella* is a significant public health concern in one-health

**Peer Reviewer** Nobuo Arai, Nogyo Shokuhin Sangyo Gijutsu Sogo Kenkyu Kiko Dobutsu Eisei Kenkyu Bumon, Tsukuba, Ibaraki, Japan

Address correspondence to Hidemasa Izumiya, izumiya@niid.go.jp.

The authors declare no conflict of interest.

See the funding table on p. 9.

settings. The multidrug-resistant (MDR) *S.* Typhimurium definitive phage type DT104, which caused a pandemic in the 1990s, is a notable example. This clone carries the *Salmonella* genomic island-1 (SGI-1) in the chromosome, which confers resistance to ampicillin, chloramphenicol, streptomycin, sulfonamide, and tetracycline (ACSSuT) (3, 4). More recently, the emergence of azithromycin resistance has become a new public health issue (5–7).

*S.* Blockley is a relatively uncommon serovar, with less than 0.05 cases per 100,000 population in the United States in recent years, compared with approximately 3 cases of *S.* Enteritidis (8). In Taiwan, *S.* Blockley accounted for 0.3% (106/40,595) of *Salmonella* isolates collected from 2004 to 2022 (9). In Japan, *S.* Blockley accounted for 1.1% (10/934) of *Salmonella* isolates reported in 2015–2016 (https://www.niid.go.jp/niid/images/iasr/archive/2016/bac/salm1516.pdf). Despite its low incidence, *S.* Blockley has garnered attention due to its association with antimicrobial resistance. Azithromycin resistance has been sporadically reported in *S.* Blockley, with *Salmonella* azithromycin resistance genomic island (SARGI) believed to play a significant role in the transmission of azithromycin resistance (10–12). However, comprehensive genomic analyses of *S.* Blockley isolates on a global scale are limited. Therefore, this study aims to perform a genomic analysis of *S.* Blockley isolates from diverse geographic regions to elucidate their phylogenetic relationships and genomic characteristics including antimicrobial resistance with a specific emphasis on azithromycin resistance.

## MATERIALS AND METHODS

### Genomic sequence data

The study incorporated genomic sequence data from 264 *S.* Blockley isolates, with 34 sequences generated for this research, 229 obtained from the public domain via Enterobase (https://enterobase.warwick.ac.uk/species/index/senterica), and 1 complete *S.* Blockley genome (accession number CP043662) sourced from the NCBI database (https://www.ncbi.nlm.nih.gov/). These isolates were collected across 33 countries in four regions, encompassing Africa, the Americas, Asia, and Europe. The isolates were collected from 1955 to 2023 (Table S1). Among the isolates, 60% (159/264) originated from human host, while 31% (82/264) were from non-human sources including poultry (17%) and livestock (7%). The source for 9% (23/264) of the isolates remained unidentified.

### Whole-genome sequencing and analysis

Whole-genome sequencing was conducted on 34 isolates using the Illumina sequencing platform (San Diego, CA, US). DNA extraction was performed using the DNeasy Blood and Tissue Kit (Qiagen, Hilden, Germany) following the manufacturer's protocol. Sequencing runs were executed on the Illumina MiSeq and/or the iSeq platforms. The obtained read sequences were assembled using the SPAdes v3.15.5 with the "—careful" option (13). To confirm the serovar Blockley, the assembled sequences (contigs) were subjected to *in-silico* serotyping using SeqSero2 v1.2.1 (14). Sequence type and eBurst Group were identified according to the scheme of Achtman (15). Antimicrobial resistance genes and plasmid replicon types were identified using ResFinder v4.1.11 (16) and PlasmidFinder (17). The *rop* gene, encoding the Rop family of plasmid primer RNA-binding protein found in the ColE1 plasmid, was identified based on the sequence under accession no. CP100731.1. Subsequently, assembled sequences (contigs) were annotated using DFAST v1.2.18 to identify genes such as *mphR(A)* and *mrx(A)* (18). The distribution of resistance and replicon genes were visualized using ggplot2 v3.4.2 (19), ComplexHeatmap v2.10.0 (20), and genoPlotR v0.8.11 (21).

### Phylogenetic analysis

A phylogenetic analysis based on single-nucleotide polymorphisms (SNPs) was conducted using genome sequencing data. Initially, snippy v4.6.0 (https://github.com/

tseemann/snippy) was employed to generate core SNP profiles, using *S*. Blockley strain 159838 (accession number CP043662) as the reference. Repetitive regions within the reference genome were identified and excluded using MUMmer v4.0.0rc1. Following extraction with Gubbins v3.3.0 and snp-sites v2.5.1 (https://github.com/sanger-pathogens/snp-sites) (22, 23), a total of 2,492 non-recombinant SNP sites were utilized to construct a phylogenetic tree. The maximum likelihood phylogeny was inferred using RAxML-NG v1.2.0 with 1,000 bootstrap replicates (24). The resulting tree was then visualized using ggtree v3.2.1 in R (25). Values of cumulative bases in recombination were obtained in the analysis of Gubbins.

## Bayesian Analysis of Population Structure

Fastbaps v1.0.8 was employed to identify Bayesian Analysis of Population Structure (BAPS) clusters in the *S*. Blockley phylogeny with "optimise.baps" prior (26). Multiple-level analysis was conducted to identify level 2 clusters for BAPS F.

## Phylogenetic dating

To estimate the divergence time of the lineage R isolates, a Bayesian dating analysis was performed using Bactdating v1.1.1 (27) with core genome single-nucleotide polymorphism (cgSNP) profiles of isolates.

## RESULTS

### Genomic characteristics and BAPS clusters

Based on the publicly available MLST scheme (15), the 264 isolates were assigned to the eBurst Group (eBG) 151 and three sequence types (STs) including ST52 ($n$ = 259), ST4928 ($n$ = 4), and ST3427 ($n$ = 1). Subsequent analysis classified the isolates into 10 distinct BAPS clusters (Fig. 1). Notably, ST4928 isolates were exclusively classified under BAPS A, while ST3427 belonged to BAPS F (F.6 among the level 2 clusters). BAPSs A–F exhibited closer genetic proximity, falling within lineage R. In contrast, BAPSs G–J clustered within another lineage, lineage S (Fig. 1). All lineage R isolates were characterized by the presence of the azithromycin resistance gene *mph(A)*. In contrast, none of the lineage S isolates harbored *mph(A)*. Lineage R displayed a low value of cumulative bases in recombination, while lineage S exhibited a high recombination frequency. The genetic clusters displayed unique geographical distribution patterns, with BAPS B, BAPS D, and BAPS E predominantly found in China, Taiwan, and Japan, respectively, while BAPS H and BAPS I were exclusively detected in the United States (Table 1). BAPS A was identified in isolates from Asia and Europe, while BAPS G was confined to European isolates. BAPSs C and J were prevalent in the Americas and Europe. BAPS F was the most widespread; it was found in isolates from Africa, the Americas, Asia, and Europe. Three BAPS F isolates were positioned outside the main cluster of BAPS F in the phylogenetic tree (Fig. 1). The discrepancy may be due to differences in the analytical methods, which can yield varying clustering results even when using the same SNP data. This phenomenon has been observed with some frequency in BAPS clustering (26). A multiple-level analysis using fastbaps was then applied to BAPS F, resulting in six subclusters, F.1–F.6. The three outlier BAPS F isolates were assigned to BAPSs F.1 and F.6, with each subcluster forming a distinct branch in the phylogenetic tree. BAPSs F.2, F.4, and F.5 were specific to Europe, while BAPS F.3 was most prevalent in Africa (Table S2).

### Antimicrobial resistance genetic determinants

*S*. Blockley isolates harbored some acquired antimicrobial resistance genes (ARGs) and mutations on *gyrA* and *parC*, conferring resistance to aminoglycosides, macrolides, tetracyclines, quinolones, phenicols, β-lactams, trimethoprim, sulfonamides, and colistin (Table 2). Notably, all isolates carried an *aac(6ʹ)-Iaa*, a silent gene that may not contribute to resistance in the *Salmonella* isolates. Resistance determinants were predominantly

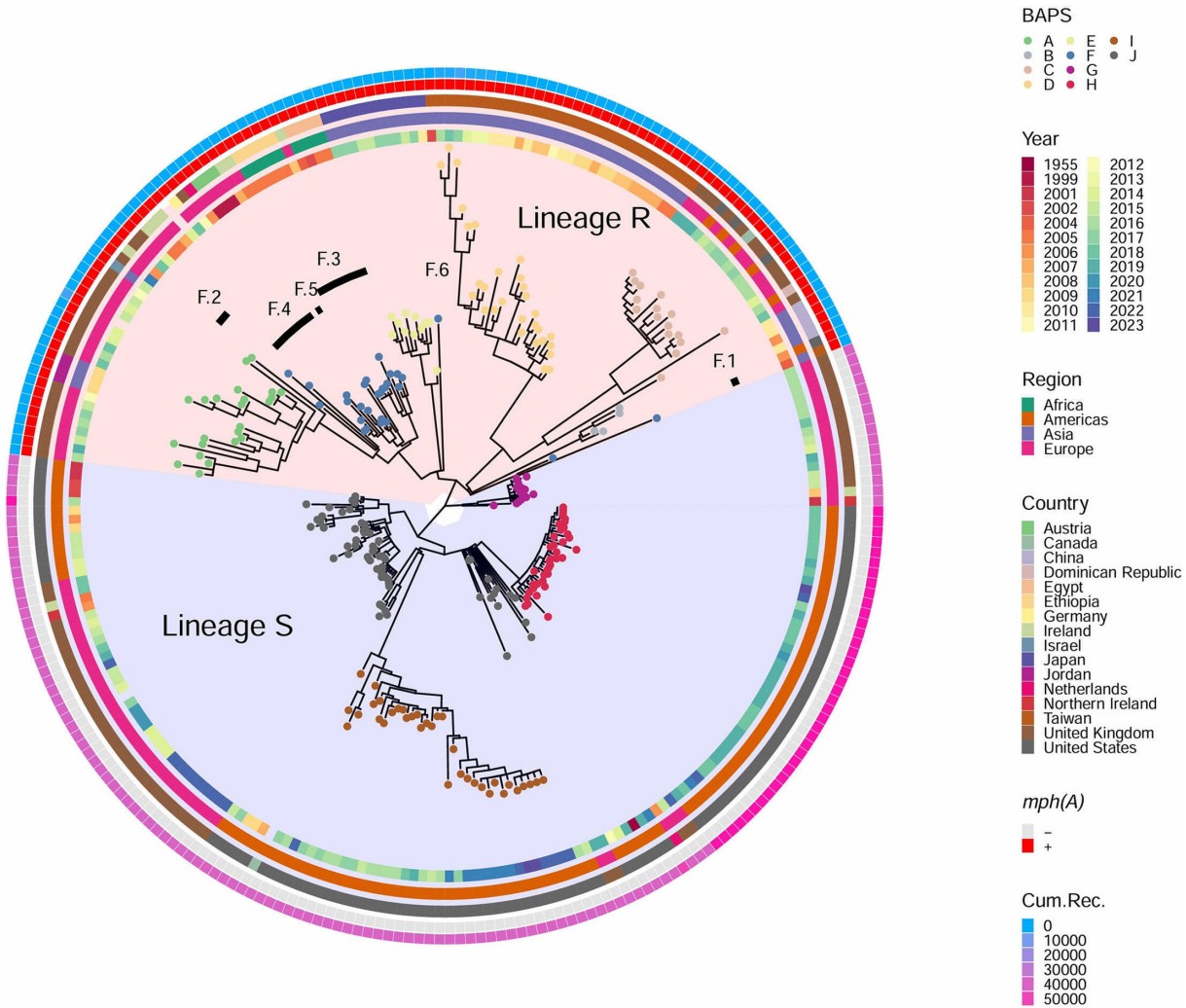

**FIG 1** Phylogenetic tree of 264 *S.* Blockley isolates based on 2,492 core SNP sites. Tips are colored by BAPS cluster according to the inset legend. Circles indicate isolation year, geographic region, country, *mph(A)*, and the values of Gubbins' cumulative bases in recombination from inner to outer. Lineages R and S are highlighted in the background. Subclusters within BAPS F are indicated by black bars.

observed in isolates of lineage R, while BAPS H and BAPS I isolates within lineage S lacked known ARGs but the silent gene *aac(6′)-Iaa*. Lineage R isolates commonly harbored *aph(3″)-Ib*, *aph(6)-Id*, *aph(3′)-Ia*, *mph(A)*, and *tet(A)*. Mutations in *gyrA* were identified exclusively in lineage R isolates, and the mutations exhibited variations within each BAPS cluster. BAPS A exhibited S83F and D87G mutations, BAPS B exhibited S83Y mutation, BAPSs C and D exhibited S83F mutation, and BAPS F exhibited S83F, D87Y (specific to BAPS F.2), or D87G (specific to BAPS F.4 and BAPS F.5) mutation, while BAPS E and lineage S (BAPSs G–J) had no mutation in *gyrA*. Certain ARGs were geographically specific, such as *floR* in isolates from Taiwan and *bla*CTX-M-15 exclusively from Japan.

## Distribution of plasmids

Seven types of replicons were identified. IncN was the most notable as it was the most abundant and found in only lineage R isolates, specifically in BAPSs A, D, E, and F (Table S3). ColE1/rop was the second most frequently identified in this study with 94% (48/51) in lineage R isolates. Of these, 75% (36/48) were detected in isolates from Taiwan (*n* = 25) and Japan (*n* = 11). As shown in Fig. S1, the *qnrB19* gene was exclusively linked to the Col(pHAD28) replicon, and the *floR* gene was strongly associated with the IncX4 replicon,

**TABLE 1** Geographic distribution of BAPS clusters[b]

| Region/country | BAPS | | | | | | | | | | Total |
|---|---|---|---|---|---|---|---|---|---|---|---|
| | A | B | C | D | E | F | G | H | I | J | |
| Africa | | | | | | **9** | | | | | **9** |
| Egypt | | | | | | 4 | | | | | 4 |
| Ethiopia | | | | | | 5 | | | | | 5 |
| Americas | | | **5** | | | **1** | | **38** | **35** | **27** | **106** |
| Canada | | | 1 | | | | | | | 1 | 2 |
| Dominican Republic | | | 1 | | | | | | | | 1 |
| United States | | | 3 | | | 1 | | 38 | 35 | 26 | 103 |
| Asia | **4** | **4** | | **29** | **11** | **2** | | | | | **50** |
| China | | 4 | | | | | | | | | 4 |
| Israel | 1 | | | | | | | | | | 1 |
| Japan | | | | | 11 | | | | | | 11 |
| Jordan | 3 | | | | | | | | | | 3 |
| Taiwan | | | | 29 | | 2 | | | | | 31 |
| Europe | **19** | | **11** | | | **15** | **16** | | | **37** | **98** |
| Austria | | | | | | 3 | | | | | 3 |
| Germany | | | | | | 1 | | | | | 1 |
| Ireland | | | | | | 7 | 1 | | | 1 | 9 |
| Netherlands | | | | | | 1 | | | | 1 | 2 |
| Northern Ireland | | | | | | | 1 | | | 1 | 2 |
| United Kingdom | 19 | | 11 | | | 3 | 14 | | | 34 | 81 |
| NA[a] | | | | | | 1 | | | | | 1 |
| Total | 23 | 4 | 16 | 29 | 11 | 28 | 16 | 38 | 35 | 64 | 264 |

[a]NA, not available.
[b]Subtotals by regions are shown in boldface type.

found primarily in Taiwanese isolates. The genes *aph(3′)-Ia*, *mph(A)*, *aph(3″)-Ib*, *aph(6)-Id*, and *tet(A)* were linked to IS*6100*, which is a component of the SARGI.

## SARGI core unit

We included *mrx(A)*, *mphR(A)*, and IS*6100* in the analysis, as these elements, along with *mph(A)*, comprise the SARGI core unit (IS*26-mph(A)-mrx(A)-mphR(A)*-IS*6100*) (10). The heatmap shown in Fig. S1 demonstrated a strong link between IS*6100* and *mrx(A)*, *mph(A), and mphR(A)*, suggesting the presence of the SARGI. Genomic analysis indicated that the SARGI was inserted into the *rbsK* gene, which encodes a protein with two functional domains, a DeoR family transcriptional regulator and a PkfB family carbohydrate kinase (ribokinase) (6, 10, 12). As shown in Fig. S2a, the inserted segments in *rbsK* comprised SARGI and an additional IS*26* composite transposon carrying *aph(3′)-Ia*, *aph(6)-Id*, and *aph(3″)-Ib*. The junction point of IS*6100* in *rbsK* was conserved, while the junction points adjacent to IS*26* were varied (Fig. S2b). Most IS*26* junction points were identified at position 953 (bp) of *rbsK*, distributed in isolates from all BAPS clusters involved. The remaining junction points were predominantly identified in BAPS F isolates; the junction at 806 was specific to BAPS F.3, while those at 929 and 939 were specific to BAPS F.4 (Table S4).

## Evolution of BAPS clusters

The emergence of the most recent ancestor of lineage R was estimated using BactDating (Fig. 2). Initial inference using a root-to-tip analysis of BactDating revealed a discrepancy in the statistical evaluation between lineages S and R. Specifically, the $R^2$ values were 0.09 and 0.62 for lineages S and R, respectively (Fig. S3), indicating that lineage R isolates could fit the analysis, whereas lineage S did not. Consequently, we applied BactDating to lineage R and a subset of lineage S, finding that the combination of lineage R and BAPS I

**TABLE 2** Distribution of resistance genes across BAPS clusters

| Class of antimicrobials | Resistance gene | BAPS A (N = 23) | BAPS B (N = 4) | BAPS C (N = 16) | BAPS D (N = 29) | BAPS E (N = 11) | BAPS F (N = 28) | BAPS G (N = 16) | BAPS H (N = 38) | BAPS I (N = 35) | BAPS J (N = 64) | Total (N = 264) |
|---|---|---|---|---|---|---|---|---|---|---|---|---|
| Aminoglycosides | aac(6')-Iaa | 23 | 4 | 16 | 29 | 11 | 28 | 16 | 38 | 35 | 64 | 264 |
| | aph(3")-Ib | 19 | 4 | 16 | 29 | 11 | 28 | 2 | | | 3 | 112 |
| | aph(6)-Id | 19 | 4 | 16 | 29 | 11 | 28 | 2 | | | 3 | 112 |
| | aph(3')-Ia | 19 | 4 | 16 | 29 | 11 | 27 | | | | | 106 |
| | aadA1 | | | | | | | 2 | | | | 2 |
| | aadA2b | | | | | | | 2 | | | | 2 |
| Macrolides | mph(A) | 23 | 4 | 16 | 29 | 11 | 28 | | | | | 111 |
| Tetracyclines | tet(A) | 23 | 4 | 16 | 29 | 9 | 26 | 4 | | | | 111 |
| Quinolones | gyrA | 23 | 4 | 16 | 29 | | 27 | | | | | 99 |
| | S83F | | | 16 | 29 | | 12 | | | | | 57 |
| | S83F+D87G | 23 | | | | | | | | | | 23 |
| | D87Y | | | | | | 12 | | | | | 12 |
| | S83Y | | 4 | | | | | | | | | 4 |
| | D87G | | | | | | 3 | | | | | 3 |
| | parC_S80R | 19 | | | | | | | | | | 19 |
| | qnrB19 | | 3 | | | | | | | | 8 | 11 |
| Phenicols | catA2 | | 4 | | 17 | 11 | 5 | | | | | 37 |
| | floR | | | | 18 | | | | | | | 18 |
| | cmlA1 | | | | | | | 2 | | | | 2 |
| Beta-lactams | blaCTX-M-15 | | | | | 10 | | | | | | 10 |
| | blaTEM-1B | | 3 | | | | 3 | 2 | | | 1 | 9 |
| | blaCMY-2 | | | 1 | | | | | | | | 1 |
| | blaSHV-12 | | | | | 1 | | | | | | 1 |
| Trimethoprim | dfrA14 | | | | | | | 2 | | 3 | | 5 |
| Sulfonamides | sul2 | | | | | | | 2 | | 3 | | 5 |
| | sul3 | | | | | | | 2 | | | | 2 |
| Polymyxin (colistin) | mcr-1.2 | | | | | 1 | | | | | | 1 |

isolates produced a moderate $R^2$ value of 0.53. Therefore, we subjected the combination of lineage R (BAPS A–F) and BAPS I to the analysis, using $10^6$ MCMC iterations. As a result, the average point mutation rate ($\mu$) was estimated at 2.71 (95% CI [CI], 2.38–3.09) substitutions per year, and the root date for the most likely ancestor root was 1979.48 [1973.14–1984.57]. All branches of the BAPS clusters of lineage R were inferred to have diverged between the 1980s and mid-1990s. Within lineage R, five types of gyrA mutations were identified (Table 2), estimated to have diverged between the 1990s and mid-2000s.

## DISCUSSION

*S*. Blockley, although a minor serovar, presents significant public health concerns due to its antimicrobial resistance, particularly to azithromycin (10, 11, 28, 29). This study provides a comprehensive genomic analysis of 264 *S*. Blockley isolates from diverse geographical regions, showing the population structure and antimicrobial resistance patterns with a specific emphasis on azithromycin resistance.

Our findings reveal that *S*. Blockley isolates in this study belong to eBG151, comprising 10 distinct BAPS clusters. These clusters can be divided into two major lineages, lineage R, which includes six BAPSs (A–F), and lineage S, which includes four BAPSs (G–J). Lineage R is characterized by the carriage of the azithromycin resistance gene *mph(A)*, a key determinant of azithromycin resistance (30), whereas lineage S lacks this gene. This genetic distinction suggests that lineages R and S have different evolutionary paths and antimicrobial resistance profiles.

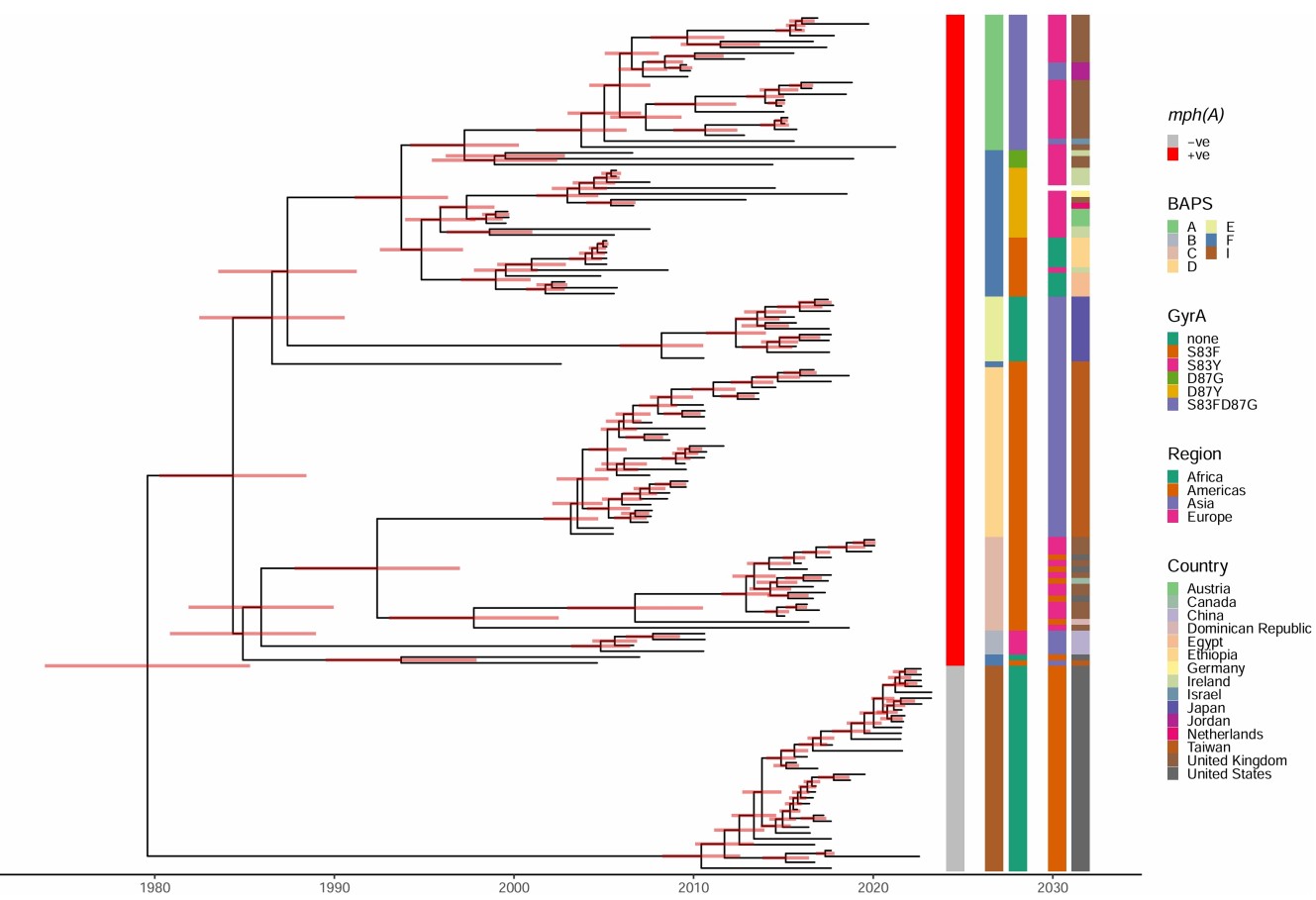

**FIG 2** The dated phylogenetic tree of 111 lineage R isolates. The tree was inferred using BactDating. The red bars indicate the 95% CI of dating.

The geographical distribution of the genetic lineages is noteworthy. Lineage R isolates were predominantly found in Asia and Africa, with specific BAPS clusters showing country-specific distributions, for instance, BAPS B in China, BAPS D in Taiwan, and BAPS E in Japan. Lineage S, on the other hand, was primarily found in isolates from Europe and the Americas, with BAPS H and BAPS I being exclusively found in those from the United States. This geographical skew highlights the regional dissemination patterns of the two lineages.

The SARGI core unit (IS*26-mph(A)-mrx(A)-mphR(A)*-IS*6100*), a key element conferring azithromycin resistance, was consistently identified in lineage R (Fig. S2). We searched the core unit in the NCBI database and found that it has been identified in numerous Enterobacteriaceae species, including *Citrobacter* spp., *Cronobacter* sp., *Edwardsiella* spp., *Enterobacter* spp., *Escherichia* spp., *Klebsiella* spp., *Kluyvera cryocrescens*, *Leclercia adecarboxylata*, *Morganella morganii*, *Proteus mirabilis*, *Providencia rettgeri*, *Raoultella* spp., *Serratia marcescens*, and *Shigella* spp., as well as in species from other families, such as *Aeromonas* spp., *Bordetella trematum*, *Shewanella* spp., and *Vibrio* spp. (data not shown). However, the genomic islands containing the core SARGI unit, *tet*(A), *aph*(3′)-*Ia*, *aph*(6)-*Id*, and *aph*(3″)-*Ib*, as shown in Fig. S2a, are found only in *S*. Haardt and *S*. Blockley.

Genomic analysis indicated that the SARGI core unit, along with IS*26*–flanking segments carrying *tet*(A), *aph*(3′)-*Ia*, *aph*(6)-*Id*, and *aph*(3″)-*Ib*, was inserted in the *rbsK* gene on the chromosome, forming a more extensive SARGI-containing genomic island (Fig. S2a). This region, comprising IS*26*, *tet*, *aph* genes, and IS*26*, is currently unique to *S*. Blockley. Chromosomal integration is an effective strategy for the dissemination of ARGs. The SARGI-containing genomic island likely contributes to the stable maintenance of resistance traits in lineage R.

Intriguingly, the junction point of IS*6100* in *rbsK* is conserved, but multiple junction points are observed for IS*26*. At the junction of IS*6100* in the *rbsK* gene, a "GG" sequence, the conserved end of the IS*6* family (31), and a target-like inverted repeat sequence "GG CTCTGTTGCAAA" (32) are found. On the other hand, the multiple junction points of IS*26* suggest that IS*26* is active in mediating intramolecular transposition that leads to DNA deletion (33).

In addition, multiple IS*26* are found in SARGI-containing genomic islands, which can mediate combinatorial variation of ARGs such as *tet(A)* and *aph(3′)-Ia–aph(6)-Id–aph(3″)-Ib* as well as mobile gene elements like IncN *rep* within the genomic island. Such variant islands have been identified in *S*. Typhimurium, *S*. Agona, and *S*. Concord (3, 34–36).

The dating analysis indicates that lineage R may have diverged around the 1980s, with most BAPS clusters within this lineage diverging before the mid-1990s. Chinese isolates (BAPS B) could have diverged earliest from lineage S. This temporal divergence correlates with the emergence of multiple *gyrA* mutation types, which occur in the chromosome and confer resistance to quinolones. These mutations were acquired in a BAPS cluster-specific manner, further illustrating the microevolutionary dynamics within lineage R.

## Conclusion

This study provides a detailed genomic analysis of *S*. Blockley, highlighting significant antimicrobial resistance patterns and population structure. *S*. Blockley falls into two main lineages, R and S, with lineage R predominantly found in Asia and Africa and characterized by the carriage of *mph(A)*, whereas lineage S, primarily found in Europe and the Americas, lacked this resistance gene. In addition to *mph(A)*, the lineage R isolates harbor genomic islands containing multiple resistance genes, including *aph(3′)-Ia*, *aph(3″)-Ib*, *aph(6)-Id*, and *tet(A)*, posing a serious public health threat. Phylogenetic analysis suggests that the lineage R diverged in the 1980s, with notable microevolutionary changes since then. Although uncommon, *S*. Blockley shows significant antimicrobial resistance, particularly to azithromycin, necessitating continuous surveillance and evaluation of its impact on public health.

## ACKNOWLEDGMENTS

We thank Ms. K. Mori and Ms. A. Takemoto for their technical assistance.

This research was supported, in part, by the Japan Agency for Medical Research and Development (AMED) under grant numbers JP24fk0108663 and JP24fk0108683.

H.I., C.-S.C., and M.O. designed the study. C.-S.C., T.S., A.N., T.H., and H.I. selected and provided isolates. H.I., M.M., C.-S.C., Y.A., and M.O. analyzed and/or interpreted the data. H.I. and C.-S.C. wrote the manuscript. All authors contributed to manuscript editing.

## AUTHOR AFFILIATIONS

[1]Department of Bacteriology I, National Institute of Infectious Diseases, Shinjuku, Tokyo, Japan
[2]Center for Diagnostics and Vaccine Development, Centres for Disease Control, Taichung, Taiwan
[3]Japan Microbiological Laboratory Co. Ltd., Miyagi, Japan
[4]Division of Bacteriology, Osaka Institute of Public Health, Osaka, Japan

## AUTHOR ORCIDs

Hidemasa Izumiya  http://orcid.org/0000-0002-9797-7402
Chien-Shun Chiou  http://orcid.org/0000-0003-3674-1030
Tetsuya Harada  http://orcid.org/0000-0001-8757-1436

## FUNDING

| Funder | Grant(s) | Author(s) |
|---|---|---|
| Japan Agency for Medical Research and Development (AMED) | JP24fk0108663,JP24fk0108683 | Hidemasa Izumiya<br>Yukihiro Akeda |

## AUTHOR CONTRIBUTIONS

Hidemasa Izumiya, Conceptualization, Formal analysis, Investigation, Visualization, Writing – original draft | Chien-Shun Chiou, Conceptualization, Formal analysis, Investigation, Project administration, Writing – original draft | Masatomo Morita, Data curation, Formal analysis, Methodology, Writing – review and editing | Toshio Sato, Investigation, Writing – review and editing | Akio Noguchi, Investigation, Writing – review and editing | Tetsuya Harada, Investigation, Writing – review and editing | Yukihiro Akeda, Conceptualization, Formal analysis, Funding acquisition, Project administration, Supervision, Writing – review and editing | Makoto Ohnishi, Conceptualization, Formal analysis, Project administration, Supervision, Writing – review and editing

## DATA AVAILABILITY

The short-read sequence data from this study have been submitted to NCBI/ENA/DDBJ under BioProject PRJDB18357 (BioSample SAMD00797488–SAMD00797521, accession numbers DRR576566–DRR576599). The accession numbers for all isolates used in this study are listed in Table S1.

## ADDITIONAL FILES

The following material is available online.

### Supplemental Material

**Supplemental figures (Spectrum02048-24-s0001.pdf).** Fig. S1 to S3.
**Supplemental tables (Spectrum02048-24-s0002.xlsx).** Tables S1 to S4.

### Open Peer Review

**PEER REVIEW HISTORY (review-history.pdf).** An accounting of the reviewer comments and feedback.

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
