## [Reviewer comments · Microbiology Spectrum]

Microbiology Spectrum

Genomic characteristics of *Salmonella enterica* serovar Blockley

Hidemasa Izumiya, Chien-Shun Chiou, Masatomo Morita, Toshio Sato, Akio Noguchi, Tetsuya Harada, Yukihiro Akeda, and Makoto Ohnishi

Corresponding Author(s): Hidemasa Izumiya, Kokuritsu Kansensho Kenkyujo

Review Timeline:

Submission Date:	August 14, 2024
Editorial Decision:	September 11, 2024
Revision Received:	October 16, 2024
Accepted:	October 25, 2024

Editor: Salina Parveen

Reviewer(s): Disclosure of reviewer identity is with reference to reviewer comments included in decision letter(s). The following individuals involved in review of your submission have agreed to reveal their identity: Nobuo Arai (Reviewer #2)

Transaction Report:

DOI: <https://doi.org/10.1128/spectrum.02048-24>

Re: Spectrum02048-24 (Genomic characteristics of *Salmonella enterica* serovar Blockley)

Dear Dr. Hidemasa Izumiya:

Thank you for the privilege of reviewing your work. Below you will find my comments, instructions from the Spectrum editorial office, and the reviewer comments.

Revision Guidelines

Sincerely,
Salina Parveen
Editor
Microbiology Spectrum

Reviewer #1 (Comments for the Author):

The manuscript titled, "Genomic characteristics of *Salmonella enterica* serovar Blockley," presents the data of phylogenetic relation, genomic characteristics, and antimicrobial resistance patterns of 264 *S. Blockley* strains isolated from diverse geographical regions. This paper is well written, seems methodologically sound and provides useful insights into characteristic of *S. Blockley*. However, the background and objectives of this study may be inadequately described. In addition, my recommendation is to analyze the region surrounding *mphA* in *mphA*-positive strains and factors related to azithromycin

resistance other than the *mphA* gene (such as *erm* genes, *mef(C)-mph(G)*, and *rpIDV* mutation) and include the results and discussion.

Reviewer #2 (Public repository details (Required)):

In this study, the authors performed whole genome sequencing for 34 *Salmonella* Blockley isolates. Therefore, they need to deposit the data in a public database. At present, the accession numbers were not described in the manuscript.

Reviewer #2 (Comments for the Author):

In the manuscript "Genomic characteristics of *Salmonella enterica* serovar Blockley"; the authors explored an extensive genetic characteristics of this serovar based on high-resolution phylogenetic analysis. This study provided the informative contents about the genetic diversity, profile of antimicrobial resistance genes, and micro-evolution of *Salmonella* Blockley that has been a public health concern.

Major comments

(i) To further understand the phylogeny of *Salmonella* Blockley, the authors should perform the BactDating for lineage R, together with lineage S. It will help to clearly show the acquisition time of SARGI and the divergence time between both lineages.

(ii) In Fig 1, the three isolates of BAPS8, the two of three are next to BAPS2 and an isolate is between BAPS4 and BAPS5, are clearly discriminated from main cluster of BAPS8. When the phylogenetic tree is inferred based on the appropriate informative SNPs, BAPS clustering should reflect the tree shape. Thus, the authors should explain the reason in the manuscript. Additionally, there is an isolate not belonging to BAPS8.4 and 8.5 between these subclusters. The authors should discuss this point too.

(iii) The authors should describe the accession numbers of all isolates used in this study. Otherwise, no one reproduce and verify the results.

(iv) If possible, the authors should re-designate the BAPS numbers, for example lineage S: BAPS1 to BAPS6, and lineage R: BAPS7 to BAPS10. It is difficult for readers to understand intuitively at present.

(v) Regarding to SARGI, the authors should consider and describe the more detailed acquisition and formation process. For example, are there the similar structures among the other bacterial species, even in partial?

Minor comments

Fig 1: *mphA* should be *mph(A)* and italic.

Table S1: The authors should clearly describe which isolates were sequenced in this study.

Figure S2a: The authors should illustrate or describe the mean of the color gradations between each strain. Probably, they indicate the identity between the nucleotide sequences.

Reviewer #1

1. The manuscript titled, "Genomic characteristics of *Salmonella enterica* serovar Blockley," presents the data of phylogenetic relation, genomic characteristics, and antimicrobial resistance patterns of 264 *S. Blockley* strains isolated from diverse geographical regions. This paper is well written, seems methodologically sound and provides useful insights into characteristic of *S. Blockley*.

However, the background and objectives of this study may be inadequately described.

Reply: Thank you for your valuable comments. We have revised the final section of the Introduction to provide a clearer explanation of the study's background and objectives. We hope the revisions meet your expectations.

2. In addition, my recommendation is to analyze the region surrounding *mphA* in *mphA*-positive strains and factors related to azithromycin resistance other than the *mphA* gene (such as *erm* genes, *mef(C)-mph(G)*, and *rplD* mutation) and include the results and discussion.

Reply: Thank you for your recommendation. We investigated the regions surrounding *mphA* in *mphA*-positive strains, and searched for additional factors related to azithromycin resistance, including *erm* genes, *mef(C)-mph(G)*, and *rplD/rplV* mutations, as suggested. However, we did not identify any additional genes or mutations. Therefore, no changes were made to the manuscript in this regard.

Reviewer #2

1. In this study, the authors performed whole genome sequencing for 34 *Salmonella Blockley* isolates. Therefore, they need to deposit the data in a public database. At present, the accession numbers were not described in the manuscript.

Reply: Thank you for bringing this to our attention. We have deposited the whole genome sequencing data in a public database, and the WGS data have been accessible. The accession numbers for all isolates used in this study have been included in Table S1.

2. In the manuscript "Genomic characteristics of *Salmonella enterica* serovar Blockley"; the authors explored an extensive genetic characteristics of this serovar based on high-resolution phylogenetic analysis. This study provided the informative contents about the genetic diversity, profile of antimicrobial resistance genes, and micro-evolution of *Salmonella* Blockley that has been a public health concern.

Reply: Thank you for your positive feedback and appreciation of our study.

3. To further understand the phylogeny of *Salmonella* Blockley, the authors should perform the BactDating for lineage R, together with lineage S. It will help to clearly show the acquisition time of SARGI and the divergence time between both lineages.

Reply: Thank you for your comments. When we applied BactDating to all isolates, the linearity of the data was insufficient ($R^2 < 0.1$). Therefore, after extensive testing, we included only the BAPS I isolates of lineage S along with the isolates of lineage R in our analysis. This combination yielded a moderate R^2 of 0.53, providing more reliable results. The updated results are presented in Figure 2 and Figure S3.

4. In Fig 1, the three isolates of BAPS8, the two of three are next to BAPS2 and an isolate is between BAPS4 and BAPS5, are clearly discriminated from main cluster of BAPS8. When the phylogenetic tree is inferred based on the appropriate informative SNPs, BAPS clustering should reflect the tree shape. Thus, the authors should explain the reason in the manuscript. Additionally, there is an isolate not belonging to BAPS8.4 and 8.5 between these subclusters. The authors should discuss this point too.

Reply: Thank you for your comments. We have addressed and explained this inconsistency in the Results section according to your suggestions.

Regarding the gap between BAPS F.4 (previously labeled as BAPS8.4) and F.5 (8.5), it is a result of the visualization in the figure. There are no isolates present between these subclusters. We apologize for any confusion this may have caused.

5. The authors should describe the accession numbers of all isolates used in this study. Otherwise, no one reproduce and verify the results.

Reply: Thank you for your comments. We have added the accession numbers for all isolates used in this study in Table S1.

6. If possible, the authors should re-designate the BAPS numbers, for example lineage S: BAPS1 to BAPS6, and lineage R: BAPS7 to BAPS10. It is difficult for readers to understand intuitively at present.

Reply: Thank you for your valuable suggestion. We have re-designated the BAPS clusters as suggested: lineage R is now labeled BAPS A to BAPS F, and lineage S is labeled BAPS G to BAPS J for clearer interpretation.

7. Regarding to SARGI, the authors should consider and describe the more detailed acquisition and formation process. For example, are there the similar structures among the other bacterial species, even in partial?

Reply: Thank you for your comments. We searched the NCBI database for SARGI-containing genomic islands and found that the genomic islands identified in this study are present only in *S. Blockley* and *S. Haardt*. In contrast, the SARGI core unit (IS26-*mph(A)*-*mrx(A)*-*mphR(A)*-IS6100) is widely distributed among various *Enterobacteriaceae* species and other bacterial families. We have incorporated these findings into the Discussion section.

8. Fig 1: *mphA* should be *mph(A)* and italic.

Reply: Thank you for pointing that out. We have corrected it to *mph(A)* and italicized it accordingly.

9. Table S1: The authors should clearly describe which isolates were sequenced

in this study.

Reply: Thank you for your suggestion. We have updated Table S1 to indicate which isolates were sequenced in this study.

10. Figure S2a: The authors should illustrate or describe the mean of the color gradations between each strain. Probably, they indicate the identity between the nucleotide sequences.

Reply: Thank you for your comments. We have updated Figure S2a to include a description of the color gradations, indicating the identity between nucleotide sequences.

Re: Spectrum02048-24R1 (Genomic characteristics of *Salmonella enterica* serovar Blockley)

Dear Dr. Hidemasa Izumiya:

Your manuscript has been accepted, and I am forwarding it to the ASM production staff for publication. Your paper will first be checked to make sure all elements meet the technical requirements. ASM staff will contact you if anything needs to be revised before copyediting and production can begin. Otherwise, you will be notified when your proofs are ready to be viewed.

Sincerely,
Salina Parveen
Editor
Microbiology Spectrum